# Effects of McConnell and Kinesio Tapings on Pain and Gait Parameters during Stair Ambulation in Patients with Patellofemoral Pain Syndrome

**DOI:** 10.3390/medicina58091219

**Published:** 2022-09-04

**Authors:** Samwon Yoon, Hohee Son

**Affiliations:** Department of Physical Therapy, College of Health Sciences, Catholic University of Pusan, Busan 46252, Korea

**Keywords:** PFPS, McConnell taping, Kinesio taping, gait parameters

## Abstract

*Background and Objectives*: The purpose of this study was to investigate the effects of McConnell and Kinesio tapings on knee pain and gait parameters during stair ambulation in patients with patellofemoral pain syndrome (PFPS). *Materials and Methods*: We selected 52 young adults suffering from anterior knee pain due to PFPS to participate. Then, we randomly assigned 26 patients to either the McConnell or the Kinesio taping groups. We measured their knee pain and gait parameters during stair ambulation before and after the interventions. For the measured data, we performed a paired *t*-test to evaluate the amount of change before and after the intervention within the groups and an independent *t*-test to compare the groups. *Results*: From the comparison within the groups, we found a significant difference in both groups in the anterior knee pain scale score (*p* < 0.05) and a significant difference between the groups as well (*p* < 0.05). As a result of the analysis of the gait parameters while ascending stairs in the comparison within the groups, both groups showed significant differences in all gait variables, except for the double-support stance (*p* < 0.05), and we found significant differences in all gait variables, except for the double-support stance, in the comparison between the groups (*p* < 0.05). Regarding the gait variables during stair descent in the comparison within the groups (*p* < 0.05), both groups showed significant differences in all of the gait variables; we noted significant differences in the double-support stance, step length, velocity, and cadence in the comparison between the groups (*p* < 0.05). *Conclusions*: The McConnell and Kinesio tapings were effective in improving knee pain and gait parameters during ambulation in patients with PFPS, but we found that the McConnell taping had a significant impact on pain reduction during stair ambulation, resulting in further improvement in the gait variables.

## 1. Introduction

The patellofemoral pain syndrome (PFPS) is the most common cause of knee joint pain [1]. An annual PFPS prevalence of 23% in adults and 29% in adolescents was reported within the general population [2]. PFPS frequently occurs in athletes, with a prevalence of 9% [3]. The pain in PFPS is located in and around the anterior region of the patella [4]. This pain is caused by repeated knee flexion and extension movements, which occur during, for example, ascending and descending stairs, running, sitting for a long time, and kneeling [5]. Various factors cause PFPS [6], including the following: structural and kinematic problems, such as imbalance between the vastus medialis (VM) and vastus lateralis (VL) muscles; misalignment of the patella; excessive increase in the Q angle; hypertonicity of the soft tissue around the patella. An abnormal patella alignment may worsen the knee kinematics, and if the activity of the VM muscle is less than that of the VL muscle or the muscle onset time is delayed, the patella glide movement may be abnormal and cause anterior knee pain. In this condition, PFPS may be exacerbated by activities such as stair ambulation, squats, and kneeling [7]. Stair ambulation can cause increased pain for patients with PFPS. Taping can be used as an intervention to aid patients with PFPS in stair ambulation. Among the various taping techniques, many researchers have studied the application of the McConnell and Kinesio tapings [8,9,10,11,12]. McConnell taping is a technique used to correct the patellar alignment with the help of inelastic tape [13]. The McConnell taping aligns the knee joint correctly by pulling the patella inward [14]. The Kinesio taping can be used to modulate muscle activation using elastic tape. The tape can be stretched by 130–140%, allowing the muscle to stretch to its maximum range of motion during movement [15]. The McConnell and Kinesio tapings are effective in reducing pain when climbing stairs [16,17], but research is needed to determine the kind of change they induce that results in pain reduction. In addition, studies on the effects of the two tapings while descending stairs are lacking. Therefore, in this study, we aimed to investigate the effects of the McConnell and Kinesio tapings on pain and gait parameters during stair ambulation in patients with PFPS.

## 2. Materials and Methods

### 2.1. Participants

We selected 52 adults who were suffering from anterior knee pain due to PFPS for this study. We informed them about the purpose and necessity of this study, and they agreed to voluntarily participate. We used the G*Power program to calculate the sample size. We applied the results obtained from the preliminary experiment conducted on 10 participants to the G*Power program, and we set the effect size, statistical power, and alpha level to 0.85, 0.8, and 0.05, respectively. As a result of the analysis, we calculated the sample size to be 46, so we selected 52 participants, considering a dropout rate of 15%. The participants were recruited by a single physical therapist, who had more than 5 years of clinical experience in knee rehabilitation. We randomly assigned the selected participants to the McConnell or the Kinesio taping groups. We performed randomization for the block of all participants via a computer random-number generator. This study was approved by the IRB committee of the Catholic University of Pusan (CUPIRB-2021-024).

The inclusion criteria were as follows: (1) a person who experienced anterior knee pain at least once within the last three months in two or more movements during prolonged sitting, stair walking, squatting, running, kneeling, and jumping [18]; (2) unilateral PFPS; (3) voluntary participation. The exclusion criteria were as follows: (1) history of subluxation or dislocation of the patella; (2) lower extremity (LE) surgery within the last year; (3) sensory or motor paralysis due to neurological damage; (4) a score of 80 or higher on the anterior knee pain scale [19].

### 2.2. Measurement Device

#### 2.2.1. Anterior Knee Pain Scale

The Kujala anterior knee pain scale (AKPS) consists of 12 items [20]. Each item consists of questions related to activities of daily living, which include limping, weight-bearing, walking, running, sitting and getting up, jumping, stair ambulation, pain, atrophy of the thigh muscles, edema, abnormal pain when moving the patella, and bending of the knee joint. The lower the score, the worse the symptoms; the better the condition, the closer to a score of 100 points. The reliability of the test is r = 0.95 [21].

#### 2.2.2. Motion Analysis

We used a 3-dimensional fully wireless motion analyzer (Myomotion System, Noraxon Co., Scottsdale, AZ, USA) for the kinematic analysis during stair walking. We set the sampling rate to 100 Hz. We processed the collected data using the MR3 software (MR3 program, Noraxon Co., Scottsdale, AZ, USA). According to the manual, we attached seven sensors as follows: to the center of the sacrum, to the center of the lateral aspect of both femurs, to the center of the lateral aspect of both calves, and to the dorsal part of both feet (Figure 1) [22]. The intraclass correlation coefficient (ICC) of the test was 0.89 [23].

### 2.3. Intervention

#### 2.3.1. McConnell Taping

For the McConnell taping, we used inelastic tape fabricated for medical purposes (ZONAS ^®^ Athletic tape, JOHNSON & JOHNSON, New Brunswick, NJ, USA). Before the taping, we cleaned the skin with alcohol. To prevent skin irritation, we attached hypoallergenic tape (Fixomull ^®^ stretch, Beiersdorf Australia Ltd., Sydney, Australia) just below the nonelastic tape [24]. We firmly attached the inelastic tape starting from the lateral side of the patella so that the patella slid inward. To induce a patella realignment, we applied sufficient force to the inside of the patella and attached the tape until skin folds appeared on the inside of the knee (Figure 2) [25].

#### 2.3.2. Kinesio Taping

We performed the Kinesio taping using elastic tape (Kinesio Tape ^®^, Atex medical, Seoul, Korea). Before the taping, we cleaned the skin with alcohol. We applied the taping in the 30° hip joint and 50° knee joint flexion supine position. As shown in Figure 3, we cut 5 cm of wide tape in a Y-shape and started taping from the medial lip of the linea aspera located at the origin of the VM, which we attached to the medial patellar tendon, which can be regarded as an insertion of the VM [26].

### 2.4. Experimental Procedure

After demonstrating the stair ambulation process to accustom the participants to the experiment, a practice was conducted for 5 min. In this study, we used a staircase with a width of 900 mm, a tread of 260 mm, and a raiser of 180 mm [27]. We asked the participants to stand at a distance of 1.5 m from where they could walk two steps to the stairs and climb the first step with the uninjured leg [28]. We measured the gait variables and the anterior knee pain scale scores during the stair walking before and after the intervention. The participants ascended and descended ten steps, and we extracted data from six of the ten steps, excluding the first and the last two steps. We recorded the measurements three times, and the participants had a 5-minute break between the measurements.

### 2.5. Gait Analysis

We used a 3D fully wireless motion analyzer to analyze the gait variables during the stair walking. We measured the stance phase, swing phase, step length, cadence, single-support stance, double-support stance, velocity, and step time during the stair walking. We processed all of the collected data using the MR3 software (MR3 program, Noraxon Co., Scottsdale, AZ, USA).

### 2.6. Analysis Method

To analyze the data that we collected in the study, we used the SPSS 28.0 for Windows. We assessed the normality of the general characteristics of all participants using the Shapiro–Wilk test. We used an independent t-test for intergroup homogeneity. We conducted a paired t-test to compare the anterior knee pain scale and the gait variables before and after the intervention within the group and an independent t-test to compare the amount of change between the groups before and after the intervention. We set the statistical significance level (α) for all the data to 0.05.

## 3. Results

### 3.1. Participant Characteristics

Table 1 presents the general characteristics of the 52 participants in this study. The average age, height, and weight were 26.9 years, 172.1 cm, and 70.2 kg, respectively. The general characteristics between the McConnell and the Kinesio taping groups were not significantly different (*p* > 0.05).

### 3.2. Changes in the Anterior Knee Pain Scale

Table 2 shows the anterior knee pain scale changes in the McConnell and the Kinesio taping groups. In the comparison within the groups, we found a statistically significant difference in the anterior knee pain scale scores for both the McConnell and the Kinesio taping groups before and after the intervention (*p* < 0.05). In the comparison between the groups, the change in the anterior knee pain scale score before and after the intervention was more significant in the McConnell tapping group than in the Kinesio taping group (*p* < 0.05).

### 3.3. Changes in the Gait Parameters

#### 3.3.1. Changes in the Gait Parameters While Ascending Stairs

Table 3 shows the gait variable changes for the McConnell and the Kinesio taping groups when climbing stairs.

#### 3.3.2. Change in the Gait Parameters While Descending Stairs

Table 4 shows the gait variable changes for the McConnell and the Kinesio taping groups when descending stairs.

## 4. Discussion

The results of our study showed that the anterior knee pain scale scores significantly increased after the intervention in both the McConnell and the Kinesio taping groups. In the comparison between the groups, those in the McConnell taping group showed a more significant increase than those in the Kinesio taping group.

Cowan et al. [28] applied the McConnell taping to 10 patients with PFPS and 12 healthy people in the same manner as in this study and found that the pain experienced by those in the McConnell taping group significantly decreased compared with that in the placebo group. Lee et al. [29] applied the Kinesio taping to the gluteus maximus muscle in 15 patients with PFPS and found that knee pain was significantly reduced during stair ambulation after the intervention. Hanafy et al. [30] assessed the stair climbing abilities of 30 women with degenerative knee arthritis after applying McConnell taping; the visual analog scale score significantly decreased in the McConnell taping group compared with that in the placebo group and the non-taping group. They also reported that McConnell taping was effective in reducing pain in patients with PFPS and knee osteoarthritis. McConnell taping can induce the patella to move along the correct path during stair ambulation without dislodging within the intercondylar sulcus. This alignment of the patella is considered effective in reducing pain because it reduces the load on the patellofemoral joint and induces kinematic benefits in the knee joint. Kinesio taping appears to be effective in reducing pain because it expands the space between the skin and fascia when muscle contraction and relaxation are repeated, leading to increased blood supply and lymphatic circulation [29,31,32]. Lee et al. [29] stated that the reason for the significant pain reduction in the Kinesio taping group is that Kinesio taping can affect the alignment of the patella. Additionally, Cowan et al. [31] reported that a misalignment of the patellofemoral joint in patients with PFPS is the main cause of pain. Based on the findings of these previous studies, alignment of the patella has a substantial effect on knee pain reduction. McConnell taping is thought to be more effective in reducing pain than Kinesio taping because it directly moves the patella in the correct path without affecting the intercondylar sulcus during stair ambulation.

As a result of the analysis of the gait variables, we found that the stance phase, single-support stance, velocity, and cadence significantly increased after the intervention in both the McConnell and the Kinesio taping groups when climbing stairs; the swing phase, step length, and step time significantly decreased. We found the same results during stair descent. The double-support stance significantly increased while descending stairs, but there was no significant change during stair ascent. Stair ambulation, which includes ascending and descending stairs, requires increased ability than level walking [33,34]. In patients with PFPS, the walking speed is slower and the movement of the LE joints is reduced due to knee pain during stair walking [30]. Power et al. [35] reported that the reason for the slow stair walking speed in patients with PFPS was to reduce the load on the patellofemoral joint. Hanafy et al. (2014) [30] reported that stair walking speed increased when McConnell taping was applied. Furthermore, Jung et al. (2008) [36] studied the effect of Kinesio taping on gait parameters in level gait and found no significant change in the stance and swing phases after they applied the Kinesio taping, but the velocity and cadence significantly increased. As such, based on the results of previous studies similar to ours, McConnell and Kinesio taping appear to be effective in improving the stair walking speed of patients with PFPS. The reason for the lengthened stance phase in this study seems to be the reduction in knee pain during weight-bearing due to the McConnell and Kinesio tapings. Cho et al. (2012) [37] reported that the stance phase of the site subjected to total knee arthroplasty was longer than that of the contralateral side, and this was the result of reduced weight-bearing at the operated site. As the weight-bearing on the knee joint increases, the stance phase increases, and the stance phase of the contralateral leg is relatively lengthened to reduce the weight-bearing. Therefore, in this study, the stance phase did not decrease in the weight-bearing section due to the pain reduction, and the stance phase was apparently sufficiently achieved.

The gait parameters significantly improved in the McConnell taping group compared with those in the Kinesio taping group because the McConnell taping had an effect on pain reduction and weight-bearing on the knee joint during the stance phase, which is considered a psychological result of knee stability [30,38]. A patellar dislocation at the patellofemoral joint during stair ambulation causes kinematic changes in the knee joint and increases the load on the joint [31]. McConnell taping can more effectively prevent patellar dislocations at the patellofemoral joint, which can reduce the load on the joint during the stance phase, resulting in a significant change in the stance phase. Since the McConnell taping directly corrects the position of the patella, it affects the kinematics of the knee joint more strongly than the Kinesio taping. In addition, according to Jung et al. (2008) [36], the stance and swing phases did not significantly change during stair walking with the Kinesio taping applied; McConnell taping was more effective at affecting the stair gait variables. 

A limitation of this study was that because we tested only the immediate effect of the intervention, we could not confirm the effect of persistent therapy due to the short duration of the intervention. A second limitation was that, because we considered only the pain and gait variables as the measurement variables, a study with additional kinematics-related measurement variables is needed for an in-depth analysis of the effect of the two tapings on stair ambulation.

## 5. Conclusions

Within the limits of this study, we drew the following conclusions: During stair ambulation, the pain and gait parameters improved in both the McConnell and the Kinesio taping groups. In the comparison between the groups, those in the McConnell taping group showed a significant change compared with those in the Kinesio taping group. Based on the above results, we found that both the McConnell and the Kinesio tapings were effective in improving the knee pain and gait parameters during ambulation in patients with PFPS, and the McConnell taping had a significant impact on the pain reduction during stair ambulation in PFPS patients, resulting in further improvements in the gait variables.

## Figures and Tables

**Figure 1 medicina-58-01219-f001:**
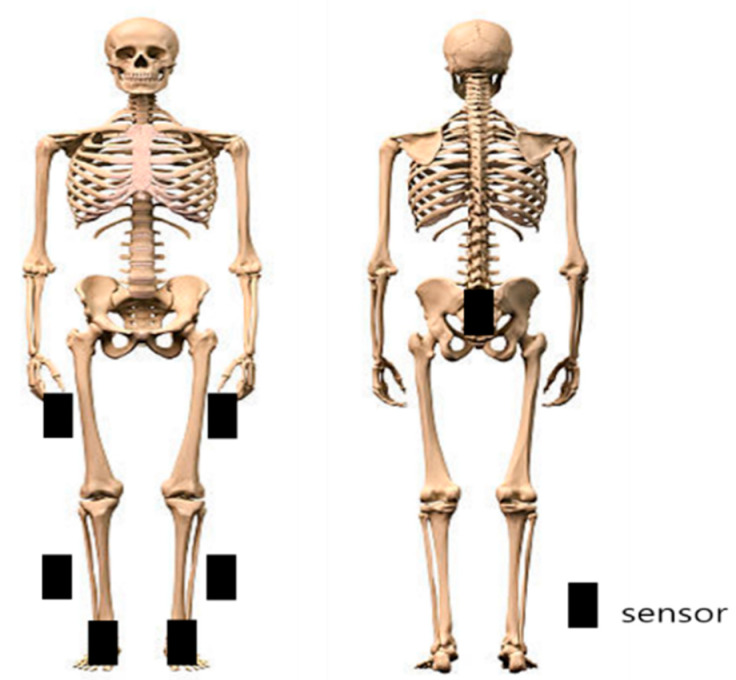
Placements of the Myomotion sensors.

**Figure 2 medicina-58-01219-f002:**
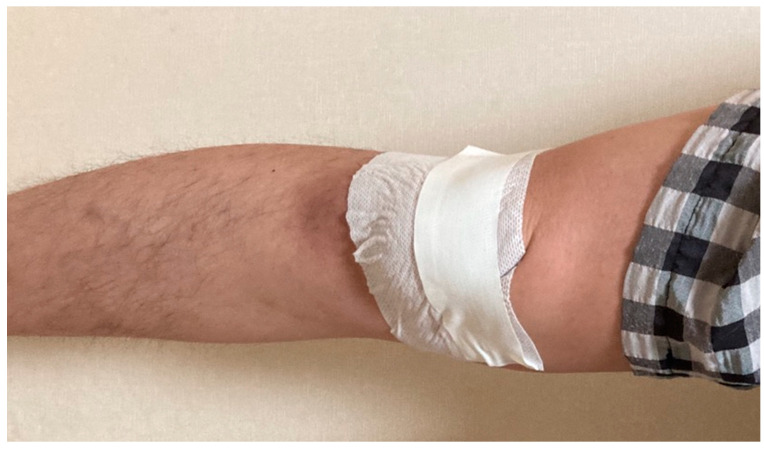
McConnell taping.

**Figure 3 medicina-58-01219-f003:**
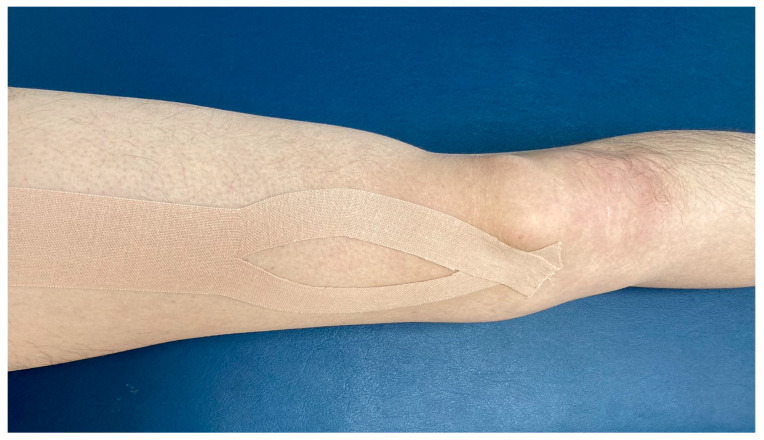
Kinesio taping.

**Table 1 medicina-58-01219-t001:** General characteristics of the study subjects.

Variable	MT (*n* = 26)	KT (*n* = 26)	*p*
Sex (male/female)	22/4	21/5	0.427
Age (years)	26.41 ± 5.36	27.13 ± 6.02	0.894
Height (cm)	171.19 ± 6.19	172.79 ± 5.83	0.847
Weight (kg)	64.75 ± 9.54	70.29 ± 11.53	0.314
AKPS (score)	65.16 ± 2.59	65.87 ± 1.85	0.206

Mean ± SD, MT, McConnell taping group; KT, Kinesio taping group; AKPS, anterior knee pain scale.

**Table 2 medicina-58-01219-t002:** Comparison of the anterior knee pain scale between the groups.

Time Point	MT (*n* = 26)	KT (*n* = 26)	*p*
Before intervention	65.16 ± 2.59	65.87 ± 1.85	0.343
After intervention	74.64 ± 4.22	70.24 ± 2.41	0.000 ^†^
*p*	0.000 *	0.000 *	

Mean ± SD, MT, McConnell taping group; KT, Kinesio taping group; * *p* < 0.05; ^†^ Significant difference between the groups (*p* < 0.05).

**Table 3 medicina-58-01219-t003:** Comparison of the gait parameters during stair ascending between the groups.

		MT (*n* = 26)	KT (*n* = 26)	*p*
S (%)	Preintervention	61.12 ± 1.75	59.88 ± 3.24	0.093
	Postintervention	69.39 ± 2.42	65.13 ± 1.93	0.001 ^†^
	*p*	0.000 *	0.000 *	
W (%)	Preintervention	38.96 ± 1.75	40.12 ± 3.25	0.126
	Postintervention	30.59 ± 2.43	34.90 ± 1.97	0.000 ^†^
	*p*	0.000 *	0.000 *	
SS (%)	Preintervention	27.88 ± 2.38	29.31 ± 3.88	0.062
	Postintervention	32.80 ± 1.61	32.22 ± 3.61	0.000 ^†^
	*p*	0.000 *	0.000 *	
DS (%)	Preintervention	35.84 ± 3.17	32.33 ± 4.21	0.000 ^†^
	Postintervention	36.10 ± 2.43	32.34 ± 4.33	0.321
	*p*	0.374	0.982	
SL (cm)	Preintervention	34.40 ± 5.70	33.68 ± 5.66	0.653
	Postintervention	27.16 ± 2.98	30.72 ± 5.32	0.000 ^†^
	*p*	0.000 *	0.000 *	
VC (km/h)	Preintervention	0.83 ± 0.19	0.88 ± 0.17	0.326
	Postintervention	1.19 ± 0.18	1.14 ± 0.18	0.000 ^†^
	*p*	0.000 *	0.000 *	
ST (ms)	Preintervention	927.52 ± 143.31	808.56 ± 158.76	0.000 ^†^
	Postintervention	608.16 ± 118.21	649.40 ± 144.20	0.000 ^†^
	*p*	0.000 *	0.000 *	0.000 ^†^
CD (step/min)	Preintervention	65.16 ± 11.18	71.64 ± 13.71	0.074
	Postintervention	95.70 ± 13.57	88.65 ± 18.39	0.000 ^†^
	*p*	0.000 *	0.000 *	

Mean ± SD, S, stance phase; W, swing phase; SS, single-support stance; DS, double-support stance; SL, step length; VC, velocity; ST, step time; CD, cadence; MT, McConnell taping group; KT, Kinesio taping group; diff, value of the difference between the pre- and post-test, * *p* < 0.05; ^†^ Significant difference between the groups (*p* < 0.05).

**Table 4 medicina-58-01219-t004:** Comparison of the gait parameters during stair descending between the groups.

		MT (*n* = 26)	KT (*n* = 26)	*p*
S (%)	Preintervention	57.22 ± 3.57	57.28 ± 3.57	0.957
	Postintervention	65.06 ± 3.25	63.96 ± 2.71	0.172
	*p*	0.000 *	0.000 *	
W (%)	Preintervention	42.77 ± 3.46	42.72 ± 2.19	0.386
	Postintervention	34.95 ± 3.17	36.04 ± 2.35	0.175
	*p*	0.000 *	0.000 *	
SS (%)	Preintervention	30.82 ± 1.94	31.35 ± 2.30	0.952
	Postintervention	34.78 ± 1.69	34.68 ± 1.77	0.251
	*p*	0.000 *	0.000 *	
DS (%)	Preintervention	23.33 ± 1.57	23.79 ± 1.73	0.321
	Postintervention	28.11 ± 2.28	26.21 ± 1.78	0.000 ^†^
	*p*	0.000 *	0.000 *	
SL (cm)	Preintervention	28.58 ± 3.48	27.88 ± 3.56	0.496
	Postintervention	32.5 ± 3.93	29.84 ± 3.22	0.000 ^†^
	*p*	0.000 *	0.000 *	
VC (km/h)	Preintervention	0.86 ± 0.17	0.88 ± 0.19	0.681
	Postintervention	1.34 ± 0.26	1.24 ± 0.24	0.000 ^†^
	*p*	0.000 *	0.000 *	
ST (ms)	Preintervention	831.88 ± 118.64	785.36 ± 145.48	0.227
	Postintervention	578.36 ± 117.97	544.20 ± 108.49	0.725
	*p*	0.000 *	0.000 *	
CD (step/min)	Preintervention	75.76 ± 9.55	78.92 ± 10.71	0.283
	Postintervention	104.01 ± 11.72	100.80 ± 9.89	0.027 ^†^
	*p*	0.000 *	0.000 *	

Mean ± SD, S, stance phase; W, swing phase; SS, single-support stance; DS, double-support stance; SL, step length; VC, velocity; ST, step time; CD, cadence; MT, McConnell taping group; KT, Kinesio taping group; diff, value of the difference between the pre- and post-test; * *p* < 0.05; ^†^ Significant difference between groups (*p* < 0.05).

## Data Availability

Not applicable.

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
