# Peer review of "Effects of McConnell and Kinesio Tapings on Pain and Gait Parameters during Stair Ambulation in Patients with Patellofemoral Pain Syndrome"

_medicina, 2022, doi:10.3390/medicina58091219_

Round 1

Reviewer 1 Report

Abstract - Spaces show up throughout - remove them.

              Line 16-19 - shorten sentence and show p values.

              Line 20 - What is double support? Define.

Intro - Line 32 - there is no verb in sentence.

           Line 38 - cause should be "causes"

           Line 45-49 - Remove sentence as it is too simple to include.

           Lin 58 - 60 - This sentence makes no sense. Rewrite. 

The Intro should make some definition of what "double support" is - define it.

M and M - Line 64 - Remove city name as it is of no consequence.

                - Need picture of Kinesio taping

                - Need picture of double support.

Table 1 - "weigh" should be "weight".

Discussion - Line 200-202 - First sentence - remove as it is redundant.

                  - Why not mention double support somewhere. What do you mean by this?

                   - There are no strengths mentioned with the Limitations. Why not? Put S and L in one paragraph. What about biases as limitations?

Conclusions - Line 273-274 - Remove as it is redundant.

                     - cut down the length of the Conclusion by 1/3 as it is redundant. Key to 2 key points.

Author Response

Comments and Suggestions for Authors

Abstract - Spaces show up throughout - remove them.

Author's note  :  I edited according to reviewer's comments.   

Line 16-19 - shorten sentence and show p values.

Author's note  :   Based on the reviewer's comments, I shortened the sentence and indicated the p-value.

Line 20 - What is double support? Define.

Author's note  :   Double support is one of the gait parameters. Gait cycle consists of two phases, where both feet are in contact with the ground, called Double Support. In order to convey the meaning of double support well, we modified it to 'double-support stance'.

- Why not mention double support somewhere. What do you mean by this?

Author's note  :   Previous studies rarely dealt with double support for PFPS patients during stair ambulation, which is thought to be because the importance was lower than that of other variables. Even so, the results should have been included in the discussion, so according to the reviewer's comments, we included the results of double support in the discussion.

Intro - Line 32 - there is no verb in sentence.

Author's note  : I put the verb 'is' in the mentioned sentence.

Line 38 - cause should be "causes"

Author's note  :  I edited according to reviewer's comments.

           Line 45-49 - Remove sentence as it is too simple to include.

Author's note  :   I removed according to reviewer's comments.

           Lin 58 - 60 - This sentence makes no sense. Rewrite. 

Author's note  :   I edited according to reviewer's comments.

M and M - Line 64 - Remove city name as it is of no consequence.

Author's note  :   I removed it according to reviewer's comments.

                - Need picture of Kinesio taping

Author's note  :   We added a picture of kinesio taping.

Table 1 - "weigh" should be "weight".

Author's note  :   I edited according to reviewer's comments.

Discussion - Line 200-202 - First sentence - remove as it is redundant.

Author's note  :  I removed it according to reviewer's comments.

                   - There are no strengths mentioned with the Limitations. Why not? Put S and L in one paragraph. What about biases as limitations?

Author's note  :  We would like to edit it after understanding clearly whether strengths mean the same as words like 'advantage' and 'merit', or 'muscular strength'. Please advise which of the two should we mention? / We tried our best to avoid bias. So there are no biases as limitations.

Conclusions - Line 273-274 - Remove as it is redundant.

Author's note  :   We edited according to reviewer's comments.

                     - cut down the length of the Conclusion by 1/3 as it is redundant. Key to 2 key points.

Author's note  :   We edited according to reviewer's comments.

Reviewer 2 Report

Review of manuscript Effects of McConell and Kinesio taping on pain and gait parameters during stair ambulation in patients with patellofemoral pain syndrome.

It is important study that aimed to evaluated effect of different methods of knee taping. Studies regarding evidence based physiotherapy are important to enable best patients treatment.

Material description: what clinical and radiological  examination did participant have to confirm diagnosis? Who evaluated the patients?

Methods: please describe methods of patients selection to each method.

Did all the patients form “McConnell” group had lateral shift of patella?

Patients were tested before and immediately after taping. How long decrease of pain and improvement of function lasted?

Conclusion are rather repetition of results, please rewrite them

Literature selection rather old. Only 2 position from last 5 years.

Minor:

Line 33 “ The prevalence of PFPS is twice as high in the 16-25 year age group” – comparing to what age group?

Line 36 “ PFPS has 37 various cause” – rather causes

Line 56 “Studies have shown…” please include citations

Figure 3 is not attached

Author Response

Material description: what clinical and radiological  examination did participant have to confirm diagnosis? Who evaluated the patients?

Author's note  :  All of the previous studies suggested a common diagnostic criteria for PFPS(Mostamand et al., 2010; Freedman et al., 2014; Rathleff et al., 2012). Therefore, our study also referred to the PFPS diagnostic criteria presented in previous studies. The patient was evaluated by a physical therapist with sufficient experience in knee rehabilitation. We added to the article who evaluated the patient.

[reference]

1) Mostamand, J., Bader, D. L., & Hudson, Z. (2010). The effect of patellar taping on joint reaction forces during squatting in subjects with Patellofemoral Pain Syndrome (PFPS). Journal of Bodywork and Movement Therapies, 14(4), 375-381.
2) Freedman, S. R., Brody, L. T., Rosenthal, M., & Wise, J. C. (2014). Short-term effects of patellar kinesio taping on pain and hop function in patients with patellofemoral pain syndrome. Sports health, 6(4), 294-300.
3) Rathleff, M. S., Roos, E. M., Olesen, J. L., & Rasmussen, S. (2012). Early intervention for adolescents with Patellofemoral Pain Syndrome-a pragmatic cluster randomised controlled trial. BMC musculoskeletal disorders, 13(1), 1-9.

Methods: please describe methods of patients selection to each method.

Author's note  :  I wrote methods of patients selection to each method in detail.

Did all the patients form “McConnell” group had lateral shift of patella?

Author's note  :  A lateral shift of patella could not be identified prior to intervention. When referring to previous studies investigating the effect of McConnell taping on PFPS, These studies did not perform a prior test for lateral shift of patella, but only performed the PFPS screening test which conducted in our study(Osorio et al., 2013; Clifford et al., 2020; Tobin & Robinson, 2000). Because we referred to the previous studies, Please understand that we did not pre-screen for lateral shift of patella.

[reference]

1) Osorio, J. A., Vairo, G. L., Rozea, G. D., Bosha, P. J., Millard, R. L., Aukerman, D. F., & Sebastianelli, W. J. (2013). The effects of two therapeutic patellofemoral taping techniques on strength, endurance, and pain responses. Physical Therapy in Sport, 14(4), 199-206.

2) Clifford, A. M., Dillon, S., Hartigan, K., O’Leary, H., & Constantinou, M. (2020). The effects of McConnell patellofemoral joint and tibial internal rotation limitation taping techniques in people with Patellofemoral pain syndrome. Gait & Posture, 82, 266-272.

3) Tobin, S., & Robinson, G. (2000). The effect of McConnell's vastus lateralis inhibition taping technique on vastus lateralis and vastus medialis obliquus activity. Physiotherapy, 86(4), 173-183.

Patients were tested before and immediately after taping. How long decrease of pain and improvement of function lasted?

Author's note  :  We apologize for not being able to confirm the continuity of treatment because we investigated the immediate effect of the interventions. 

Therefore, we wrote that further studies on the continuity of taping interventions are likely to be necessary in the limitation

.

Conclusion are rather repetition of results, please rewrite them

Author's note  :  We edited according to reviewer's comments.

Literature selection rather old. Only 2 position from last 5 years.

Author's note  :  We have replaced some older literature, including papers from the 90s, with papers published within 5 years as much as possible.

Minor:

Line 33 “ The prevalence of PFPS is twice as high in the 16-25 year age group” – comparing to what age group?

Author's note  :  It meant that it was twice as high compared to all age groups except for the 16-25 year age group, but there was insufficient evidence to prove this. Therefore, we modified it to a different sentence with solid evidence.

Line 36 “ PFPS has 37 various cause” – rather causes

Author's note  :  We edited according to reviewer's comments.

Line 56 “Studies have shown…” please include citations

Author's note  :  We included relevant  citations according to reviewer's comments.

Figure 3 is not attached

Author's note  :  We added a picture of Figure 3

Round 2

Reviewer 1 Report

Please add to the Methods about the local Ethics approval.

Was consent obtained with the use of a locally approved consent form? Please add this to the paper.

Author Response

Review 1: Please add to the Methods about the local Ethics approval.

Response 1:  We added to the Methods about the local Ethics approval.

Review 2: Was consent obtained with the use of a locally approved consent form? Please add this to the paper.

Response 2:  We added the subject consent form approved by the ethics committee to the paper. Please understand that the consent form for the subject is not in English as consent form must be written in the language(korean) appropriate for the nationality of the subject.

Reviewer 2 Report

I still have concerns regarding this manuscript: lack of radiological examination, lack of medical doctor clinical examination, only 5 year old experience of physiotherapist. No information if all the patients had lateral shift of patella. They got treatment for lateral shift correction. If they did not have shift, correction is unnecessary.

Author Response

Review 1: I still have concerns regarding this manuscript: lack of radiological examination, lack of medical doctor clinical examination, only 5 year old experience of physiotherapist.

Response 1:  There have been studies in which doctors performed PFPS diagnosis, but there were also studies in which physical therapists and even athletic trainers performed diagnostic examinations. McConnell (1986), Powers (1998), Gilleard et al (1998), McConnell & Bennell (2011), Lee & Cho (2013), Chen et al (2008) performed pfps diagnostic examination by a physical therapist. Waryas & McDermott (2008) mentioned that pfps diagnostic test can be performed not only by doctors but also by athletic trainers. In this way, there are many previous studies that a physical therapist and athletic trainer can diagnose pfps, and even in our study, a physical therapist directly conducted the examination, so please understand.

Review 2: No information if all the patients had lateral shift of patella. They got treatment for lateral shift correction. If they did not have shift, correction is unnecessary.

Response 2:  McConnell, who invented McConnell taping, said that the McConnell taping method for pfps patients is to pull inward (McConnell 1986; McConnell & Bennell, 2011).

In other studies that applied McConnell taping to pfps, they performed McConnell taping as suggested by McConnell (Powers 1998; Gilleard et al., 1998; Osorio et al, 2013; Lee & Cho 2013).

In fact, even McConnell did not argue that McConnell taping should be performed only in pfps patients with a lateral shift of patella (McConnell 1986; McConnell & Bennell, 2011),

and in four studies conducted McConnell taping on pfps were not screened for lateral shift of patella in pfps patients(McConnell 1986; Powers 1998; Gilleard et al., 1998; McConnell & Bennell, 2011).

In addition, as in our study, two studies analyzing the effects of McConnell taping and kinesio taping did not screen lateral shift of patella in pfps patients.

Our study also wanted to find out how McConnell taping and Kinegio taping, which are the most known tapings for pfps, affect pfps. Whether or not there is lateral shift, we wanted to know how the two most popular tapings among pfps taping interventions affect knee pain and gait parameters when applied to pfps patients. Since there is no lateral shift of patella among the diagnostic criteria for pfps, some pfps patients may have no lateral shift of patella. However, in our study, if the lateral shift of patella was included as the subject inclusion criterion, the experimental results could not be generalized to all pfps patients. For generalization, We wanted to know the effect of McConnell taping and Kinesio taping for pfps who selected as essential conditions without detailed conditions.

[Additional Information]

McConnell did not say that McConnell taping was only applied to prevent lateral shift of patella.

He said that the key point of McConnell taping is not to correct the lateral shift, but to ensure that the patella is constantly correctly positioned on the intercondylar sulcus when the knee joint moves (Chang et al., 2015).

Therefore, it seems that many previous studies that performed McConnell taping did not include the lateral shift of patella as a mandatory screening test.

[reference]

  1. Powers, C. M. (1998). Rehabilitation of patellofemoral joint disorders: a critical review. Journal of orthopaedic & sports physical therapy, 28(5), 345-354.
  2. McConnell JS. The management of chondromalacia patellae: a long-term solution. ilustmlian Journal ofPhysiothpraly. 1986;32:215-2'23. :
  3. Gilleard, W., McConnell, J., & Parsons, D. (1998). The effect of patellar taping on the onset of vastus medialis obliquus and vastus lateralis muscle activity in persons with patellofemoral pain. Physical therapy, 78(1), 25-32.
  4. McConnell, J., & Bennell, K. (2011). Conservative management of anterior knee pain: the McConnell program. In Anterior knee pain and patellar instability (pp. 191-208). Springer, London.
  5. Osorio, J. A., Vairo, G. L., Rozea, G. D., Bosha, P. J., Millard, R. L., Aukerman, D. F., & Sebastianelli, W. J. (2013). The effects of two therapeutic patellofemoral taping techniques on strength, endurance, and pain responses. Physical Therapy in Sport, 14(4), 199-206.
  6. Lee, S. E., & Cho, S. H. (2013). The effect of McConnell taping on vastus medialis and lateralis activity during squatting in adults with patellofemoral pain syndrome. Journal of exercise rehabilitation, 9(2), 326.
  7. Chen, P. L., Hong, W. H., Lin, C. H., & Chen, W. (2008). Biomechanics effects of kinesio taping for persons with patellofemoral pain syndrome during stair climbing. In 4th Kuala Lumpur International Conference on Biomedical Engineering 2008 (pp. 395-397). Springer, Berlin, Heidelberg.
  8. Chang, W. D., Chen, F. C., Lee, C. L., Lin, H. Y., & Lai, P. T. (2015). Effects of Kinesio taping versus McConnell taping for patellofemoral pain syndrome: a systematic review and meta-analysis. Evidence-Based Complementary and Alternative Medicine, 2015.
  9. Waryasz, G. R., & McDermott, A. Y. (2008). Patellofemoral pain syndrome (PFPS): a systematic review of anatomy and potential risk factors. Dynamic medicine, 7(1), 1-14.
